# ^177^Lu-DOTATATE Efficacy and Safety in Functioning Neuroendocrine Tumors: A Joint Analysis of Phase II Prospective Clinical Trials

**DOI:** 10.3390/cancers14246022

**Published:** 2022-12-07

**Authors:** Alberto Bongiovanni, Silvia Nicolini, Toni Ibrahim, Flavia Foca, Maddalena Sansovini, Arianna Di Paolo, Ilaria Grassi, Chiara Liverani, Chiara Calabrese, Nicoletta Ranallo, Federica Matteucci, Giovanni Paganelli, Stefano Severi

**Affiliations:** 1Osteoncology and Rare Tumors Center, IRCCS Istituto Romagnolo per lo Studio dei Tumori (IRST) “Dino Amadori”, 47014 Meldola, Italy; 2Nuclear Medicine and Radiometabolic Unit, IRCCS Istituto Romagnolo per lo Studio dei Tumori (IRST) “Dino Amadori”, 47014 Meldola, Italy; 3Osteoncologia, Sarcomi dell’Osso e dei Tessuti Molli, e Terapie Innovative, IRCCS Istituto Ortopedico Rizzoli, 40136 Bologna, Italy; 4Unit of Biostatistics and Clinical Trials, IRCCS Istituto Romagnolo per lo Studio dei Tumori (IRST) “Dino Amadori”, 47014 Meldola, Italy

**Keywords:** neuroendocrine tumors, carcinoid syndrome, PRRT, insulinoma, ^177^Lu-DOTATATE

## Abstract

**Simple Summary:**

Neuroendocrine Tumors are rare cancers with limited therapeutic options. Functioning NETs could produce bioactive peptides leading to a specific syndrome that impacts on patients’ quality of life and also on survival—F-NETs patients who are refractory to SSA respond to 177LU-PRRT with a benefit in terms of prognosis.

**Abstract:**

Introduction: Neuroendocrine tumors (NETs) are rare malignancies with different prognoses. At least 25% of metastatic patients have functioning neuroendocrine tumors (F-NETs) that secrete bioactive peptides, causing specific debilitating and occasionally life-threatening symptoms such as diarrhea and flushing. Somatostatin analogs (SSAs) are usually effective but beyond them few treatment options are available. We evaluated the clinical efficacy of 177 Lu-DOTATATE in patients with progressive metastatic F-NETs and SSA-refractory syndrome. Patients and Methods: A non-pre-planned joint analysis was conducted in patients enrolled in phase II clinical trials on metastatic NETs. We extrapolated data from F-NET patients with ≥1 refractory sign/symptom to octreotide, and ≥1 measurable lesion. Syndrome response (SR), overall survival (OS), progression-free survival (PFS), tolerance and disease response were analyzed. Results: Sixty-eight patients were enrolled, the majority (88.1%) with a SR. According to RECIST criteria, 1 (1.5%) patient showed a CR, 21 (32.3%) had a PR and 40 (61.5%) SD. At a median follow-up of 28.9 months (range 2.2–63.2) median PFS was 33.0 months (95%CI: 27.1–48.2). Median OS (mOS) had not been reached at the time of the analysis; the 2-year OS was 87.8% (95%CI: 76.1–94.1). Syndromic responders showed better survival than non-responders, with a 2-year OS of 93.9% (95%CI: 92.2–98.0) vs. 40.0% (95%CI: 6.6–73.4), respectively. A total of 233 adverse events were recorded. Grade 1–2 hematological toxicity was the most frequent. Conclusion: The 177 Lu-DOTATATE improved symptoms and disease control in patients with F-NETs. Treatment was well tolerated. The syndrome had an impact on both quality of life and OS.

## 1. Introduction

Neuroendocrine tumors (NETs) are rare slow-growing tumors arising from cells of the diffuse neuroendocrine system commonly located in the gastrointestinal (GI) tract or pancreas [1]. Although rare, NETs have long fascinated clinicians because they have the ability to secrete bioactive substances, mainly peptides and amines, leading to distinct clinical syndromes. These tumors are thus categorized as “functioning” or “non-functioning” [2,3].

Around one third of patients with GI-NETs develop diarrhea, abdominal pain and flushing, a complex clinical scenario known as carcinoid syndrome (CS) characterized by an increased release of serotonin [4]. CS symptoms include flushing (90%), diarrhea (70%), abdominal pain (40%), and rarely bronchospasm [5]. The long exposition to high serotonin levels leads to nutritional deficiencies, intestinal fibrotic changes and the development of carcinoid heart disease (CHD), characterizing a poor overall prognosis [6,7]. In fact, the tumor progression increasing the serotonin levels exacerbates a progressive right-sided heart disease, leading to cardiac cachexia [8,9]. Many of these patients (up to 70% in older studies), usually with liver metastases, developed carcinoid heart disease from the serotonin-driven development of endocardial fibrotic plaques in the heart [10].

The most common functional pancreatic NEN (panNEN), insulinoma, causes hypoglycemia which, when diagnosed early, can be successfully treated with surgical resection. However, in around 10% of the patients affected by metastatic insulinoma, the hypoglycemia conventional treatment with high intravenous glucose infusion, diazoxide and somatostatin analogs may be ineffective [11]. Other abnormal hormone production associated with gastro-entero-pancreatic (GEP) and lung NETs include glucagon, vasoactive intestinal peptide, adrenocorticotropic hormone, somatostatin and parathyroid hormone-related protein [12]. Furthermore, there are some other very rare forms of functioning NETs (f-NETs) characterized by different hormones secreted such as luteinizing hormone, renin, GLP-1, IGF-2, erythropoietin, entero-glucagon and cholecystokinin [13,14].

Several treatments for neuroendocrine tumors have been validated or investigated in prospective clinical trials focusing on the anti-proliferative effect including somatostatin analogs (SSAs), multi-kinase inhibitors such as sunitinib, axitinib lenvatinib and pazopanib and the mammalian target of rapamycin inhibitor (mTOR), Everolimus [15,16].

Tumor progression can severely impact on quality of life (QoL) and survival, underscoring the need to achieve significantly controlled hormone secretion to improve outcome [6,17]. More than 90% of G1 and G2 (G2) GEP-NETs overexpress SSTRs, in particular subtype 2, usually detected through 68gallium PET/CT or OctreoScan [18]. These imaging studies, if positive, make peptide receptor radionuclide therapy (PRRT) a potential option.

PRRT with lutetium-177-DOTA0-Tyr3]octreotate (^177^Lu-DOTATATE) plays a prominent role in the therapeutic scenario of metastatic NETs. Recently a randomized controlled trial (NETTER-1) demonstrated that ^177^Lu-DOTATATE PRRT is effective in controlling the tumor growth in patients with progressive, metastatic non-functioning small intestinal NETs [19]. Other large institutional studies including pNETs reported high overall response rates after the administration of PRRT. Most of these studies include patients with non-functioning pNETs and only a small number of patients with functioning pNETs [20,21].

Yttrium-90 (90Y) edoteotide was used in 90 patients with metastatic CS and octreotide-refractory symptoms. It produced durable responses, with a significantly longer PFS in patients who experienced a sustained improvement in diarrhea, and with an expected but acceptable adverse effect profile after three cycles of 4.4 GBq of 90Y-edoteotide every 6 weeks [22]. Another study focused on the use of 90Y-PRRT for symptom control and the authors described the case of a patient affected by insulinoma who had a normalization of glucose level after the fourth therapy administration [23]. However, no data are available on the role of ^177^Lu-DOTATATE PRRT in patients with syndromic metastatic GEP-NETs.

We thus carried out a joint analysis of two prospective phase II trials focusing on the safety and activity of ^177^Lu-DOTATATE in patients with metastatic functioning NETs of a different origin.

## 2. Materials and Methods

### 2.1. Study Design

This analysis included data from the prospective open-label phase II trial LUNET, (NCT01740427, approval date: 20 November 2013) and the LUTHREE (NCT01942135, approval date: 16 March 2016) clinical study. Each study was designed as a prospective, open-label phase II randomized trial evaluating the use of 18.5 GBq or 27.5 GBq of ^177^Lu-DOTATATE.

### 2.2. Patients

The intent-to-treat (ITT) population included all patients with functioning NETs enrolled in these 2 studies. Eligible patients had to fulfil all the following criteria: a histology-proven inoperable or metastatic well-differentiated, G1-G3 GEP-NET; a significant uptake (grade 3 or 4) according to Krenning score) [24] at somatostatin receptor imaging (SRI) with 111 In-pentetreotide or a positive 68Ga-DOTATOC PET/CT; a measurable disease; no other concomitant anti-tumoral treatments (such as chemo- or radiotherapy) received less than 4 weeks prior to ^177^Lu-DOTATATE; no bone marrow invasion ≥25%. All patients included had to be progressed to at least one previous treatment at the imaging evaluation according to RECIST (Response Evaluation Criteria in Solid Tumors) criteria [25]; no other concomitant tumors (with the exception of in situ basal cell carcinoma and radically treated cervical cancer); Eastern Cooperative Oncology Group (ECOG) performance status ≤2; life expectancy >6 months. None of the patients with carcinoid heart disease included were treated with valve replacement before PRRT.

Sixty-eight patients were considered eligible for the study. The decision to enroll patients onto PRRT protocols was taken by the multidisciplinary and interdisciplinary specialized Neuroendocrine Neoplasia and Endocrine Gland Tumors Board of our institute (IRCCS Istituto Romagnolo per lo Studio dei Tumori (IRST) “Dino Amadori”), which is a member of the European Union Reference Network for Rare Cancers Neuroendocrine Tumor Group (EURACAN G4 NET).

### 2.3. Primary and Secondary Endpoint

Carcinoid syndrome symptoms were defined as the presence of diarrhea >3 bowel movements/day, characterized by: (1) a daily stool consistency ≥5 on the Bristol Stool Form scale (1 hard lumps) to 7 (liquid) for ≥ 50% of the days; (2) average daily cutaneous flushing frequency of ≥2; (3) average daily rating of ≥3 for abdominal pain [26]. For malignant insulinoma and ACTH-producing NETs, hypoglycemia (glucose level <50 mcg/L) and ACTH (adrenocorticotropic hormone) levels over the normal range were considered signs of syndromic disease, respectively. The histological diagnosis was performed or confirmed by a dedicated pathologist on the basis of the 2017 and 2019 WHO classifications. Side-effects were evaluated and graded according to National Cancer Institute Common Toxicity Criteria (version 4.03) [27].

All patients underwent a 12 week disease evaluation with a CT scan or MRI before the start of PRRT. A clinical and biological evaluation, as well as an imaging evaluation were performed every 3 months or when clinically indicated. If a scan had been performed elsewhere, the imaging was reviewed by an expert radiologist from our NEN multidisciplinary board.

### 2.4. Outcome Measures

SR was defined as follows: clinically meaningful reduction in the frequency of bowel movements assessed at ≥30% over 12 weeks; in cases of hypoglycemia, normalization of serum glycemic level and/or suspension of use of diazoxide or glucose solution; normalization of ACTH level in patients with ACTH-producing tumors. Additional analyses of efficacy endpoints included time to syndrome response considered as time in months between treatment initiation and syndrome response.

Overall response rate (ORR) was calculated in terms of the proportion of patients with CR or PR according to RECIST v.1.1 criteria [24]. Measurable disease was a criterion for study inclusion. Patients were considered evaluable for response if at least one cycle of study drug was administered and at least one follow-up tumor evaluation imaging was performed.

OS was considered as the time between the start of PRRT and date of death, while PFS was considered as the time between start of PRRT and date of disease progression. Patients without events (death or disease progression) at the time of the analysis were censored, using the date of the most recent follow-up evaluation.

Progression Free and Overall Survival (PFFS and OS) were visualized using the Kaplan–Meier curves and compared with the log-rank test. Ninety-five percent confidence intervals (95%CI) by non-parametric methods were calculated. A *p*-value < 0.05 was considered statistically significant. All the statistical analyses were completed using STATA/MP 15.0 for Windows (Stata CorpLP, College Station, TX, USA). Safety assessments included documentation of adverse events according to the National Cancer Institute Common Terminology Criteria for Adverse Events (CTCAE, v. 4.03) [27].

### 2.5. Treatment Protocol

The 177 Lutetium was purchased from Advanced Accelerator Applications ready for use at the requested dosages in both protocols. In particular, the dosage was 3.7 GBq for the LUNET protocol (GEP-NET G1/2 with FDG-negative PET/CT) in which patients were randomized to have 5 or 7 cycles of therapy, 8 weeks apart. The LUTHREE protocol (“basket” protocol with Ki-67 <35% patients) was stratified by the presence of risk factors for renal and bone marrow toxicity. Patients at risk were treated with a 3.7 GBq repeated for 5 cycles, while not at-risk cases underwent 5.5 GBq repeated for 5 cycles. In both cases, patients were randomized to receive treatment every 5 or 8 weeks to evaluate potential differences in terms of toxicity and efficacy.

The radiopharmaceutical product was infused intravenously over 30 min by a dedicated pump system (patent US 7842023 B2; Paganelli–Chinol). The procedure has been described elsewhere [28].

### 2.6. Evaluation of Side-Effects

For each PRRT cycle, laboratory analyses were performed on the day before therapy and for the whole inpatient stay. A re-evaluation every 2 and 6 weeks after each treatment cycle was performed. Additionally, during the follow-up period, patients performed laboratory exams at 12 +/− 4 weeks. Hematological and non-hematological toxicities were evaluated according to Common Terminology Criteria for Adverse Events (CTCAE) V4.03. All the hematological and clinical exams, cumulative absorbed kidney dosage and post-treatment weight loss were recorded before and after each treatment cycle and during follow-up.

## 3. Results

Table 1 shows the principal demographics and disease characteristics of the 68 patients (male = 38 (55.9%) and female = 30 (44.1%)) with f-NETs included in the study. Median age: 65 years (range 42–81). All patients had a performance status between 0 and 2. Comorbidities and risk factors for kidney injury are reported in Appendix A. The median cumulative dosage administered was 22.2 GBq (range 3.7–32.9), and a median of five treatment cycles (range 2–7) were performed. The majority of patients had G2 NETs (*n* = 45, 67.1%) and two (3.0%) had G3 NETs. Of note, 55 (80.9%) patients had tumors of gastrointestinal origin. Diarrhea and flushing were reported by 62 patients (91.1%), one patient (1.5%) had an adrenocorticotropic hormone (ACTH)-producing gastrointestinal neuroendocrine tumor, and five (7.4%) had insulin-producing disease. Five (7.4%) had carcinoid heart disease. All patients were actively progressing at the start of treatment and had SSA-refractory syndromic disease. ^18^FDG -PET/CT was performed in 54 (80.6%) patients and was positive in 26 (48.2%) and negative in 28 (51.8%). Forty-three (63.2%) patients had undergone baseline surgical resection of the primary tumor. Almost the half of the cohort had metastatic disease at diagnosis.

All patients had received at least one prior line of systemic treatment. Prior therapies included SSAs (also high-dosage SSAs), targeted therapy, cytotoxic chemotherapy and locoregional hepatic therapy to control syndromic disease.

### 3.1. Outcomes

The overall response rate of the 65 evaluable patients was 33.8%, with a DCR of 95.3%. Only three (4.7%) patients showed disease progression according to RECIST criteria (Table 2). Fifty-nine patients had showed a syndrome response, while the 3 (42.8%) patients with RECIST PD and the 4 (57.2%) with SD or PR did not. Patients who received a cumulative dosage of >18.5 GBq were more likely to have a syndrome response (Table 3).

The mOS had not been reached at a median follow-up of 28.9 months (range 2.2–63.2). mPFS was 33 months (95%CI: 27.1–48.2) and the 2-year OS was 87.8% (95%CI: 76.1–94.1). A statistically significant difference was seen in mPFS according to PS ECOG (Appendix A). No significant differences were seen in OS or PFS in relation to gender, age (< or ≥65 years), site of primary tumor, grading and type of syndrome (Appendix A). Patients with a positive ^18^FDG-FDG-PET/CT had a similar mOS. mPFS was 38.7 months (95%CI: 17.1–not estimable (NE)) compared to 54.1 months (95%CI: 37.1–NE) for the ^18^FDG-PET/CT-negative group (Appendix A). This difference was not statistically significant (*p*-value = 0.076). Despite the short median follow up, Ki-67 value < or ≥10% seems not to have an impact on mOS but appears to have affected mPFS, i.e., 39.5 months (95%CI: 28.3–NE) and 25.3 months (95%CI: 11.4–31.2), respectively (*p*-value = 0.002) (Figure 1).

Patients who underwent previous primary tumor surgery showed a benefit in terms of both mPFS and mOS, with a mPFS of 39.2 months (95%CI 28.3–54.1) compared to 24.9 months (13.8–39.5) for those who did not undergo surgery. This difference was significant (*p*-value = 0.027). The former group showed a 2-year OS of 94.1% (95%CI 78.4–98.4) compared to 76.1% (95%CI 51.3–89.4) for the latter group (*p*-value = 0.049) (Figure 2A,B).

Median time to syndrome response was 5.0 months (95%CI 4.0–6.5). Time to best tumor response was 7.3 months (95%CI: 5.8–7.9) (Appendix A).

In 20 of 65 (30.8%) patients included the 5-hydroxyindoleacetic acid (5-HIAA value was recorded at the pre-treatment visit and after the second cycle of treatment. The median basal value was 356.4 mg/24 h (normal range 2.0–10.0). The post second cycle median value was 28.5 mg/24 h.

The five patients with insulinoma started the ^177^Lu-PRRT-treatment with diazoxide and a 24-h intravenous glucose support. All showed a normalization of blood glucose levels and were able to suspend supportive therapy (Appendix A). The patient with an ACTH-producing tumor had a clinical syndrome similar to that of patients with Cushing’s syndrome, which is also characterized by an increased levels of urinary free and serum cortisol and nephropathy due to proteinuria >3.5 g/daily and hypoalbuminemia. The normalization of ACTH, urinary free and serum cortisol levels led to an improvement in renal function.

### 3.2. Safety

Side effects were recorded for all the patients (*n* = 68). A total of 233 adverse events were recorded. Transient G1 or G2 myelotoxicity were the most frequently reported side effects (Table 4). Six (2.4%) G3 hematological toxicity events were reported: one anemia (0.4%), one leukopenia (0.4%) one thrombocytopenia (0.4%) and two lymphocytopenia (0.8%). In these patients, the treatment has been withdrawn. G1 asthenia was observed in 16 (6.9%) cases. In one patient with a malignant insulinoma, a grade 3 hypoglycemia was recorded during the first cycle but easily managed with a progressive improvement in glucose levels after the second one. No carcinoid crisis were recorded. There were no G4 toxicities.

## 4. Discussion

F-NETs are characterized by the production of active peptides that can lead to the onset of specific symptoms such as abdominal pain, diarrhea, and episodes of flushing in patients with CS, or hypoglycemia in those with a malignant insulinoma. CS is most commonly found in patients with small intestinal NETs and multiple liver metastases, and occurs less frequently in patients with lung NETs. Insulinomas are linked to pancreatic tumors. Initially, f-NET symptoms can be treated with SSAs such as lanreotide and octreotide. However, 20% to 40% of patients experience a recurrence of symptoms and >60% may experience a worsening of these symptoms even during treatment. The current limited clinical evidences and the different response to SSA have made patient management challenging. Everolimus has been found useful in malignant insulinoma because of its antiproliferative and hypoglycemia-controlling effect. The rarity of these syndromes has given little opportunity for carrying out pre-planned prospective clinical trials. However, results recently published on Telotristat revealed promising efficacy in reducing bowel movements in patients with CS [29,30].

To the best of our knowledge, ours is the first prospective large study to show the efficacy of 177Lu-PRRT in F-NETs. Several institutional prospective studies have been published on the antitumor efficacy of 177Lu-PRRT, but data on the specific outcome of F-NET patients are limited [31]. In 2014, Seregni et al. published a study on 26 mNET patients (21 with functioning tumors) treated with both 90Y- and 177Lu-PRRT, reporting non-well-defined symptom control in 90% of patients. The ORR was 33.8% [23].

In our study, 86.7% of F-NET patients with CS, malignant insulinoma or ACTH-producing neuroendocrine tumors showed a syndromic response. Furthermore, patients with malignant insulinoma treated with diazoxide and glucose infusion were able to suspend supportive therapy, with a clear benefit on quality of life.

In the phase II study published by Bushnell et al. in 2010, 90 patients with metastatic NETs were treated with 90Y-PRRT if at least one of the following symptoms was reported: diarrhea, flushing, abdominal pain, nausea/vomiting, fatigue, loss of appetite and more. Seventy-nine of the 90 patients completed a self-report questionnaire on symptoms, of whom 56 indicated diarrhea as the main symptom. mPFS and mOS were 16.3 and 26.9 months, respectively, the former significantly higher in the 38 patients who showed an improvement in diarrhea than in the 18 patients who did not (18.2 vs 7.9 months, respectively) (*p* = 0.031) [22].

We also noted this difference in our study. In particular, non-responder syndromic patients (*n* = 8, 11.9%) had a mPFS of 11.3 months (2.2–12.8) with respect to the 38.7 months (28.3-50.9) of responders. Furthermore, there was a significant difference in prognosis in these two patient categories, with a 2-year OS of 40% for the former compared to 93.9% for the latter (*n* = 59, 88.1%) (*p* < 0.001). This indicates that syndromic disease not only had an impact on quality of life but also on the natural history of the tumor. Conversely, 177Lu-PRRT had both an antiproliferative effect and an impact on symptom control. Clinical response to PRRT has, in fact, become the most important prognostic factor in the management of NETs. Interestingly, in our F-NET cohort, symptom regression preceded tumor response by around 2 months, indicating its potential usefulness to identify patients who are more likely to respond to therapy. Furthermore, in four patients obtaining a disease response, a lack of syndrome response has been recorded suggesting an independent mechanism in the resistance to therapy. This should be better investigated in dedicated clinical trials.

Another point of discussion is the cut-off dosage of 177Lu-PRRT required to impact both disease and syndrome response. Previous studies by our group investigated the minimum effective dosage of 177Lu-PRRT needed to achieve an antiproliferative effect on neuroendocrine tumors. On the basis of our results, at least 18,5 GBq of 177Lu-PRRT was considered necessary to obtain a syndrome response. This facilitates the tailoring of radionuclide receptor therapy in patients with different characteristics, such as syndromic disease, high tumor burden or presence of comorbidities [32].

It is universally accepted that surgical resection of liver metastases as part of the clinical management of NETs also plays a role in metastatic disease. As recommended in the NANETS (North American Neuroendocrine Tumor Society) and ENETS (European Neuroendocrine Tumor Society) guidelines, palliative resection of the primary tumor should be strongly considered for patients with symptoms caused by small intestinal obstruction, occlusion, or tumor bleeding in order to prevent clinical deterioration or complications that could lead to the patient’s death [33].

In CS patients, symptoms are not correlated with the mechanical impairment of the tumor but rather with the ‘functionality’ of the neuroendocrine tumor. In fact, in our cohort, those who underwent primary tumor surgery seemed to respond better to PRRT and had a better prognosis than those without, supporting the use of primary tumor surgery in patients with metastatic disease. A possible different explanation could be related to better clinical conditions of the first group of patients than the second one. Dedicated clinical studies are needed to solve this point. Our study also highlighted the activity of 177Lu-PRRT in patients with Ki-67 ≥ 10%, indicating the prognostic value of this proliferation index. Confirmation of this is eagerly awaited from the results of the ongoing NETTER-2 trial (clinical.trial.gov number: NCT03972488). The main limitation of our study was that the F-NET subgroup analysis was not pre-planned. The use of self-administered quality of life questionnaires would also have shed further light on the impact of F-NETs on daily living. Another issue is the lack of the serial evaluation of 5 hydroxy indoleacetic acid (5HIAA) in all patients with CS, as this is thought to be a useful biomarker of disease. We decided to add these data recorded only in one third of patients because of possible interest.

## 5. Conclusions

In conclusion, our findings show that ^177^Lu-PRRT is effective and safe in all patients with F-NETs. Syndrome response was an indicator of potential RECIST disease response and could be useful to gauge whether a treatment is likely to work or not. Prospective clinical trials focusing on syndrome response are needed.

## Figures and Tables

**Figure 1 cancers-14-06022-f001:**
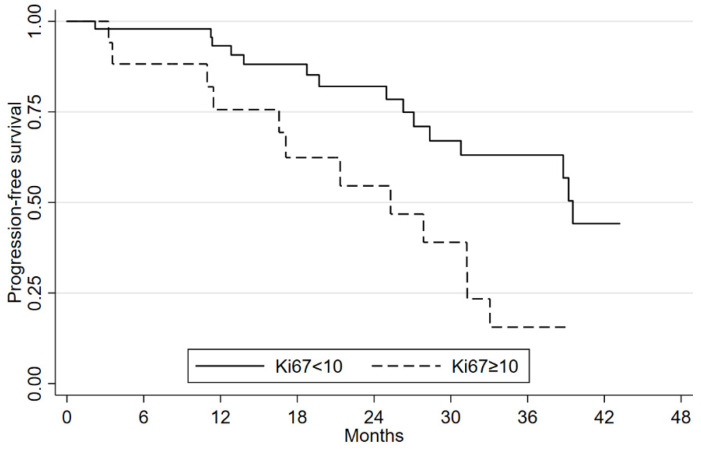
Progression-free survival according to Ki67 values.

**Figure 2 cancers-14-06022-f002:**
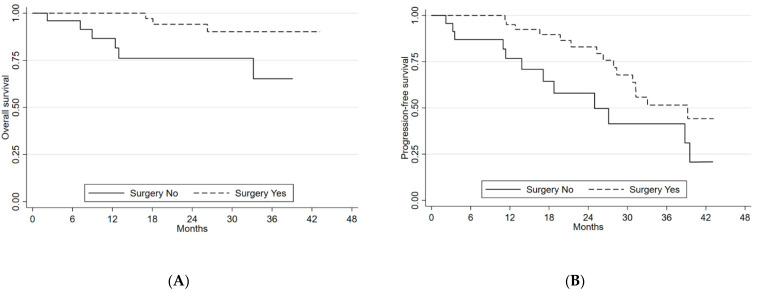
(**A**) Overall survival and (**B**) progression-free survival according to surgery.

**Table 1 cancers-14-06022-t001:** Main patient characteristics.

Median Age at Treatment, Years (Range)	65 (42–81) No. (%)
Gender	
Male	38 (55.9)
Female	30 (44.1)
Previous surgery	43 (63.2)
Presence of metastasis at diagnosis	51 (75.0)
ECOG PS	
0	52 (76.5)
1	15 (22.0)
2	1 (1.5)
Ki-67	
≤2	20 (29.9)
>2 and ≤20	45 (67.1)
>20	2 (3.0)
Unknown	1
Site of primary disease	
Lung	5 (7.7)
Pancreas	5 (7.7)
Gastrointestinal tract	55 (84.6)
Unknown	3 (4.4)
Grade	
1	18 (26.5)
2	48 (70.6)
3	2 (2.9)
Carcinoid heart disease	5 (7.4)
Somatostatin receptor imaging	
^68^Ga-PET/CTscan	60 (88.2)
OctreoScan	8 (11.8)
^18^F-FDG-PET/CT	54 (80.6)
Positive	26 (48.2)
Negative	28 (51.8)
Previous treatments	
First-line	68 (100.0)
SSA	58
Cisplatin-based chemotherapy	9
Targeted therapy	2
Capecitabine alone or in combination with oxaliplatin	2
Temozolomide alone or in combination with capecitabine	2
Second-line	24 (35.3)
SSA HD or SHIFT	9
SSA + TACE/TAE	4
Targeted therapy	4
Chemotherapy	7
Third-line	7 (10.3)
SSA HD	3
Targeted therapy	2
Chemotherapy	2
More than 3 treatment lines	5 (7.3)

ECOG PS, Eastern Cooperative Oncology Group; SSA, somatostatin analog; HD, high dose; TACE, transarterial chemoembolization; TAE, transarterial embolization.

**Table 2 cancers-14-06022-t002:** Treatment characteristics of F-NET patients treated with ^177^Lu-PRRT.

	No. (%)
Concomitant SSA	
No.	4 (5.9)
Sandostatin	37 (58.7)
Lanreotide acetate	26 (41.3)
Unknown	1
Best response to treatment	
Complete response	1 (1.5)
Partial response	21 (32. 3)
Stable disease	40 (61.5)
Progressive disease	3 (4.7)
Not evaluable	3
Syndrome response	
Yes	59 (88.1)
No	8 (11.9)
Unknown	1
Median no. cycles (range)	5 (2–7)
Median cumulative activity (range)(GBq)	22.2 (3.7–32.9)

SSA, somatostatin analog.

**Table 3 cancers-14-06022-t003:** Relation between best response to treatment and response to syndrome.

	Response to Syndrome		Overall
	No (%)	Yes (%)	No. (%)
Best response to treatment			
Complete or partial response	2 (28.6)	20 (34.5)	22 (33.8)
Stable disease	2 (28.6)	38 (65.5)	40 (61.5)
Progressive disease	3 (42.8)	0 (0.0)	3 (4.7)
Unknown	1	1	2
Cumulative activity (GBq)			
≤18.5	7 (87.5)	23 (39.0)	30 (44.8)
>18.5	1 (12.5)	36 (61.0)	37 (55.2)

**Table 4 cancers-14-06022-t004:** Hematological and non-hematological toxicity in patients (*n* = 62) with functioning NETs treated with ^177^Lu-PRRT.

Adverse Event	Grade 1 No. (%)	Grade 2 No. (%)	Grade 3 No. (%)	Grade 4 No. (%)	Total No. (%)
Hematological toxicity					
Anemia	15 (6.4)	5 (2.1)	1 (0.4)	-	21 (9.0)
Leukopenia	26 (11.1)	15 (6.4)	1 (0.4)	-	42 (17.9)
Neutropenia	8 (3.4)	12 (5.1)	1 (0.4)	-	21 (8.9)
Lymphocytopenia	2 (0.8)	-	2 (0.8)	-	4 (1.6)
Thrombocytopenia	14 (6.0)	2 (0.8)	1 (0.4)	-	17 (7.2)
Non-hematological toxicity					
Alopecia	1 (0.4)	-	-	-	1 (0.4)
Asthenia	16 (6.9)	1 (0.4)	1 (0.4)	-	18 (7.7)
Constipation	-	2 (0.8)	1 (0.4)	-	3 (1.2)
Cough	-	1 (0.4)	-	-	1(0.4)
Diarrhea	4 (1.7)	5 (2.1)	1 (0.4)	-	10(4.3)
Dizziness	2 (0.8)	-	-	-	2 (0.8)
Dyspnea	-	1 (0.4)	-	-	1 (0.4)
Fever	1 (0.8)	-	-	-	1 (0.8)
Headache	1 (0.8)	3 (1.2)	-	-	4 (1.7)
Liver toxicity	1 (0.4)	-	-	-	1 (0.4)
Muscle weakness	2 (0.8)	-	-	-	2 (0.8)
Nausea	10 (4.3)	5 (2.1)	-	-	15 (6.4)
Pain	2 (0.8)	4 (1.7)	-	-	6 (2.5)
Itch	1 (0.4)	1 (0.4)	-	-	2 (0.8)
Skin rash	3 (1.2)	4 (1.7)	-	-	7 (2.9)
Weight gain	1 (0.4)	-	-	-	1 (0.4)
ALT increase	5 (2.1)	-	-	-	5 (2.1)
Lipase increase	-	1 (0.4)	-	-	1 (0.4)
GGT increase	1 (0.4)	5 (2.1)	1 (0.4)	-	7 (3.0)
Creatinine increase	2 (0.8)	1 (0.4)	-	-	
Other symptoms	22 (9.4)	9 (3.9)	5 (2.1)	-	56 (24.0)
Total	140 (60.1)	78 (33.5)	15 (6.4)	-	233 (100.0)

Other symptoms: Abdominal pain; bone pain; increased bilirubin; dyspepsia; herpes labialis; hypoglycemic crisis; flushing; lower limb pain; tachycardia; steatorrhea; fatty liver. ALT, alanine aminotransferase; GGT, gamma-glutamyl transferase.

## Data Availability

The datasets generated and/or analyzed during the current study are available from the corresponding author on reasonable request.

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
