# Peer review of "177Lu-DOTATATE Efficacy and Safety in Functioning Neuroendocrine Tumors: A Joint Analysis of Phase II Prospective Clinical Trials"

_cancers, 2022, doi:10.3390/cancers14246022_

Round 1
Reviewer 1 Report
Bongiovanni et al. assessed the treatment efficacy of 177-Lu-DOTATATE in 68 participants with progressive metastatic functional neuroendocrine tumors from two phase II clinical trials. The majority of patients had functioning GI-NET, aka carcinoid tumors, with a few patients with other functioning NETs including 5 cases with insulinoma and one with ACTH-producing tumors. The authors reported 88% symptoms control following the treatment. The anti-tumor efficacy was comparable to previous studies with rare CR, 30% PR and 40% SD. Median OS was not reached, likely due to a short-term follow up. Median PFS was 33 months. Participants with functional NETs and Ki-67 less than 10% had longer PFS following the treatments, compared to those with Ki-67 >10%. Those who had previous surgery experienced longer PFS and OS, compared to those who did not have surgery. The treatment was well-tolerated with rare Grade 3 adverse events.
Overall, the manuscript was clearly written. The focus on symptomatic improvement is somewhat interesting. The strength is in the prospective assessment of treatment efficacy. The weaknesses included the heterogenous group of patients with inherit selection bias in prior treatments and the ability to receive >500 mci of treatment.
1. The reviewer would recommend to adjust certain statements that strongly and incorrectly suggested the absolute certainty.
a. Line 229: “Ki67% < or > 10% did not have impact on mOS”: this is probably overstated since the follow up is too short to determine the difference in the NETs that were mostly G1 and G2.
b. Line 235: “patients who underwent previous primary tumor surgery showed a benefit in mPFS and mOS”: because there was an inevitable selection bias to offer surgery to patients with advanced NET, one cannot state that surgery provided benefit without a comparable control group. Being selected as surgical candidates by itself with or without surgery is probably a favorable prognostic factor because of less advanced, resectable disease in patients with no prohibitive co-morbidities.
i. The authors should discuss the results in the context of selection bias.
2. Is there a correlation between those with CR/PR (lower tumor volume) and symptomatic control vs. SD? In other words, is the symptomatic control independent to tumor response?
3. Please expand why some patients were unable to receive > 500 mci. Was it because of adverse events?
Author Response
Reviewer response 1
Bongiovanni et al. assessed the treatment efficacy of 177-Lu-DOTATATE in 68 participants with progressive metastatic functional neuroendocrine tumors from two phase II clinical trials. The majority of patients had functioning GI-NET, aka carcinoid tumors, with a few patients with other functioning NETs including 5 cases with insulinoma and one with ACTH-producing tumors. The authors reported 88% symptoms control following the treatment. The anti-tumor efficacy was comparable to previous studies with rare CR, 30% PR and 40% SD. Median OS was not reached, likely due to a short-term follow up. Median PFS was 33 months. Participants with functional NETs and Ki-67 less than 10% had longer PFS following the treatments, compared to those with Ki-67 >10%. Those who had previous surgery experienced longer PFS and OS, compared to those who did not have surgery. The treatment was well-tolerated with rare Grade 3 adverse events.
Overall, the manuscript was clearly written. The focus on symptomatic improvement is somewhat interesting. The strength is in the prospective assessment of treatment efficacy. The weaknesses included the heterogenous group of patients with inherit selection bias in prior treatments and the ability to receive >500 mci of treatment.
Reply: We thank the reviewer for the positive comments. We modify the manuscript according the suggestion received.
- The reviewer would recommend to adjust certain statements that strongly and incorrectly suggested the absolute certainty.
- Line 229: “Ki67% < or > 10% did not have impact on mOS”: this is probably overstated since the follow up is too short to determine the difference in the NETs that were mostly G1 and G2.
- Line 235: “patients who underwent previous primary tumor surgery showed a benefit in mPFS and mOS”: because there was an inevitable selection bias to offer surgery to patients with advanced NET, one cannot state that surgery provided benefit without a comparable control group. Being selected as surgical candidates by itself with or without surgery is probably a favorable prognostic factor because of less advanced, resectable disease in patients with no prohibitive co-morbidities.
- The authors should discuss the results in the context of selection bias.
Reply: we agree with reviewer’s suggestion. We have understated the sentence. Lines 226-227. Furthermore, we have added a statement on a different explanation of this benefit due to a selection bias in the discussion. Page 10 lines 323-325.
- there a correlation between those with CR/PR (lower tumor volume) and symptomatic control vs. SD? In other words, is the symptomatic control independent to tumor response?
Reply: We would like to thank the reviewer to have the opportunity to discuss this point. Despite the low number of patients, 4 obtaining a CR/PR didn’t have a syndrome response. We add a brief consideration in the discussion section, Page 10 lines 308-311
- Please expand why some patients were unable to receive > 500 mci. Was it because of adverse events?
Reply: Some patients didn’t receive 500 mCI of 177Lu due to clinical or disease progression. Only few patients didn’t receive it due to toxicity. We specify it in the text line 256
Reviewer 2 Report
This study provides analysis of the outcomes of participants with functional NET syndromes of two phase II trials of PRRT. This is important, because a syndromic response without tumour reduction is still an important outcome for patients.
Writing is of a high quality. Line 111 "all patients were progressed" appears incorrectly worded.
Introduction
Paragraph from lines 68 to 71 appears out of place, and possibly included for self-reference?
The introduction could be shorter to allow more detail in methods and results, if the word limit is an issue.
Materials and methods
Description does not provide enough evidence of Cushing's syndrome from ectopic ACTH (presumably a p-NET?). An ACTH above normal reference intervals could occur in a patient under stress (eg. with highly symptomatic metastatic disease). Some other investigation would have occurred, - given this was for one patient, the confirmatory test could be mentioned in the result section.
Results
For a study of functional NET, outcomes directly related to the functional aspect of NET is barely described. Although numbers are low, it is worth mentioning the frequency of hypoglycaemic crisis in the text. No mention of carcinoid crisis is made, although flushing is recorded under "other" adverse events. If carcinoid crisis did not occur it is worth mentioning also.
For the five patients with carcinoid heart disease, were they pre- or post valve replacement? Was there any adverse event related to deterioration of their heart disease?
While it is likely that many did not have pre- and post treatment 5-HIAA, for those who did, including those results will be of interest to readers.
Discussion
An important limitation is that carcinoid syndrome is defined as diarrhoea with 1 of 3 features symptoms, but other causes of diarrhoea (and there are many in patients with NET), have not been ruled out. This is likely a design flaw of the original trials and cannot be altered by the authors but should be mentioned as a limitation. The diagnosis of carcinoid syndrome is not straightforward, and it is reasonable to take a pragmatic approach, however this should nonetheless be acknowledged.
Author Response
This study provides analysis of the outcomes of participants with functional NET syndromes of two phase II trials of PRRT. This is important, because a syndromic response without tumour reduction is still an important outcome for patients.
Reply: We thank the reviewer for the positive comments. We modify the text in order to improve the quality of the manuscript
Writing is of a high quality. Line 111 "all patients were progressed" appears incorrectly worded.
Reply: We thank the reviewer. We modify the text (page 3 line 108)
Introduction
Paragraph from lines 68 to 71 appears out of place, and possibly included for self-reference?
The introduction could be shorter to allow more detail in methods and results, if the word limit is an issue.
Reply We thank the reviewer. The length of the introduction is compatible with the journal We modify the reference according the suggestion received in order to avoid self-citation appearing inappropriate.
Materials and methods
Description does not provide enough evidence of Cushing's syndrome from ectopic ACTH (presumably a p-NET?). An ACTH above normal reference intervals could occur in a patient under stress (eg. with highly symptomatic metastatic disease). Some other investigation would have occurred, - given this was for one patient, the confirmatory test could be mentioned in the result section.
Reply: We agree the reviewer 2. Our intention to be as concise as possible maybe led to incomplete information. All patients were assessed by an Endocrinologist of our multidisciplinary team. In particular the one described is a rare case of gastrointestinal ACTH producing neuroendocrine tumors. We performed all the test useful for ACTH producing tumor diagnosis and to follow up. We included the information about the primary tumor and results post therapy at page 4 line 195 and page 8 lines 247-249, respectively.
Results
For a study of functional NET, outcomes directly related to the functional aspect of NET is barely described. Although numbers are low, it is worth mentioning the frequency of hypoglycaemic crisis in the text. No mention of carcinoid crisis is made, although flushing is recorded under "other" adverse events. If carcinoid crisis did not occur it is worth mentioning also.
Reply: We thank the reviewer for the positive suggestions. We add these information in the text in the safety paragraph.
For the five patients with carcinoid heart disease, were they pre- or post valve replacement? Was there any adverse event related to deterioration of their heart disease?
Reply: All the five patients were in the pre-valve replacement. We add this information at page 3 lines 113-114
While it is likely that many did not have pre- and post treatment 5-HIAA, for those who did, including those results will be of interest to readers.
Reply: in 20 patients these information was available. We add this information in the text page 8 lines (244-247)
Discussion
An important limitation is that carcinoid syndrome is defined as diarrhoea with 1 of 3 features symptoms, but other causes of diarrhoea (and there are many in patients with NET), have not been ruled out. This is likely a design flaw of the original trials and cannot be altered by the authors but should be mentioned as a limitation. The diagnosis of carcinoid syndrome is not straightforward, and it is reasonable to take a pragmatic approach, however this should nonetheless be acknowledged.
Reply: We agree with the reviewer. However we would like to be ensure that all patients were evaluated by our multidisciplinary team and a different nature of the diarrhoea was excluded (i.e. steatorrhea from SSA). We better define the carcinoid syndrome symptoms recorded for the analysis.
Reviewer 3 Report
The paper deals with the treatment effect and side effects of Lutetium-177 based PRRT in functioning NETs in 68 patients with respect to symptom control and RECIST criteria. This analysis included data from the prospective open-label phase II trial LUNET and the LUTHREE clinical study. The conclusion of the authors, derived from their results, is that PRRT is safe and syndrome response is a potential indicator for RECIST response.
In the introduction as well as in the discussion the authors focus on serotonin as the main issue concerning symptoms belonging to the NET syndrome. Therefore, it is essential to show the results of serotonin levels (also in relation to RECIST outcome) during the treatment cycles. It cannot be excluded that serotonin levels are equal indicators.
The patient group would be more homogenous if only GI tract NETs are included - particularly the one case with an ACTH-producing tumor should be excluded.
Radioactivity doses sgould be given in GBq or MBq (as done e.g. in line 102), not in mCi (e.g. line 193 and Table 3).
Minor point: the median follow-up (29 months) is rather low for NETs, which are often stable for several years. This issue should be discussed.
Author Response
The paper deals with the treatment effect and side effects of Lutetium-177 based PRRT in functioning NETs in 68 patients with respect to symptom control and RECIST criteria. This analysis included data from the prospective open-label phase II trial LUNET and the LUTHREE clinical study. The conclusion of the authors, derived from their results, is that PRRT is safe and syndrome response is a potential indicator for RECIST response.
In the introduction as well as in the discussion the authors focus on serotonin as the main issue concerning symptoms belonging to the NET syndrome. Therefore, it is essential to show the results of serotonin levels (also in relation to RECIST outcome) during the treatment cycles. It cannot be excluded that serotonin levels are equal indicators.
Reply: We thank the reviewer for the positive comments and for the opportunity to improve the manuscript. as explained in the discussion the lacking of these information is a limitation of the study. However we agree with both reviewers (2 and 3) and we add the information on pre- and post- 5HIAA evaluation available in 20 of 65 patients. Unfortunately, due to logistic problems ( the 24 hours urine collection and the importance of the 177LU administration timing) some patients performed only the baseline text and some others performed the text in an external laboratory.
The patient group would be more homogenous if only GI tract NETs are included - particularly the one case with an ACTH-producing tumor should be excluded.
Reply: We agree with the reviewer but we think that is important also to show and disseminate the activity of the treatment also in these rare cases.
Radioactivity doses sgould be given in GBq or MBq (as done e.g. in line 102), not in mCi (e.g. line 193 and Table 3).
Reply: we modify the radioactivity value in the table and in the text.
Minor point: the median follow-up (29 months) is rather low for NETs, which are often stable for several years. This issue should be discussed.
Reply: We agree with the reviewer however we noted that despite the median follow up seems to be low , it was sufficient to analyze the impact of the treatment in this subgroup of patients maybe because of a different prognosis compared with the one of NF-NET patients
Round 2
Reviewer 1 Report
The authors responded to my questions and comments.
Reviewer 3 Report
-